# NS2 induces an influenza A RNA polymerase hexamer and acts as a transcription to replication switch

Junqing Sun[1,2,3,7], Lu Kuai [ID][3,7], Lei Zhang[3,4,7], Yufeng Xie [ID][3,5], Yanfang Zhang[3], Yan Li[3], Qi Peng[3], Yuekun Shao[3], Qiuxian Yang[3], Wen-Xia Tian [ID][1✉], Junhao Zhu[3], Jianxun Qi [ID][3], Yi Shi [ID][2,3,6✉], Tao Deng [ID][3✉] & George F Gao [ID][1,2,3,6✉]

## Abstract

**Genome transcription and replication of influenza A virus (FluA), catalyzed by viral RNA polymerase (FluAPol), are delicately controlled across the virus life cycle. A switch from transcription to replication occurring at later stage of an infection is critical for progeny virion production and viral non-structural protein NS2 has been implicated in regulating the switch. However, the underlying regulatory mechanisms and the structure of NS2 remained elusive for years. Here, we determine the cryo-EM structure of the FluAPol-NS2 complex at ~3.0 Å resolution. Surprisingly, three domain-swapped NS2 dimers arrange three symmetrical FluPol dimers into a highly ordered barrel-like hexamer. Further structural and functional analyses demonstrate that NS2 binding not only hampers the interaction between FluAPol and the Pol II CTD because of steric conflicts, but also impairs FluAPol transcriptase activity by stalling it in the replicase conformation. Moreover, this is the first visualization of the full-length NS2 structure. Our findings uncover key molecular mechanisms of the FluA transcription-replication switch and have implications for the development of antivirals.**

**Keywords** Influenza A Virus; NS2; FluAPol; FluAPol-NS2 Complex; Transcription to Replication Switch
**Subject Categories** Chromatin, Transcription & Genomics; Microbiology, Virology & Host Pathogen Interaction; Structural Biology

## Introduction

Influenza viruses are classified into four types (A, B, C, and D viruses) in which influenza A virus (IAV) can cause seasonal epidemics, pandemics, and sporadic avian influenza virus infections in humans (Adams et al, 2017). Although the various types of influenza virus display differences in host range and virulence, they all have a similar life cycle.

Influenza A virus contains an eight-segmented negative-sense single-stranded RNA genome which encodes at least ten major viral proteins (PB2, PB1, PA/P3, HA, NP, NA, M1, M2, NS1, and NS2). Each RNA segment is packaged into a viral ribonucleoprotein (vRNP) complex consisting of multiple nucleoproteins (NP) and a heterotrimeric polymerase (FluPol) complex comprised of either the PA or P3, PB1, and PB2 subunits. During infection, the virus binds to a cell-surface sialic acid receptor and enters the host cell by endocytosis. Subsequently, the vRNPs are released into the cytoplasm and trafficked to the nucleus to initiate genome transcription and replication. In the early stage of infection, viral messenger RNA (mRNA) is produced that encodes viral proteins (Eisfeld et al, 2014). During transcription, vRNP is in close physical association with the host RNA polymerase II machinery (Engelhardt et al, 2005). Transcription is primed by a necessary cap-snatching process which obtains capped oligonucleotides from nascent host capped RNAs, and terminated by stuttering at the 5' proximal oligo(U) polyadenylation signal (Poon et al, 1999; Wandzik et al, 2020; Zhu et al, 2023).

Viral replication is a two-step process which occurs only after new FluPol and NP are produced and transported into the nucleus. In the first step of genome replication, positive-sense complementary RNA (cRNA) is generated by the polymerase using negative-sense vRNA in the vRNP as a template, and is assembled into a cRNP, a structure similar to vRNP. In the second step, cRNA is used as a template to produce new molecules of vRNA (Fan et al, 2019; Peng et al, 2019; York et al, 2013). The nascent RNA is captured by viral polymerase and bound by nucleoproteins to form vRNP/cRNP structures in which the RNA is stabilized and protected (Jorba et al, 2009; Vreede et al, 2011; Zhu et al, 2023). In contrast to transcription, replication initiates in a primer-independent (de novo) manner and terminates by reading through the polyadenylation signal. The process of replication, particularly cRNA to vRNA synthesis, requires

[1]College of Veterinary Medicine, Shanxi Agricultural University, Jinzhong 030801, China. [2]Shanxi Academy of Advanced Research and Innovation, Taiyuan 030032, China. [3]CAS Key Laboratory of Pathogen Microbiology and Immunology, Institute of Microbiology, Chinese Academy of Sciences, Beijing 100101, China. [4]Institute of Pediatrics, Shenzhen Children's Hospital, Shenzhen 518026, China. [5]Department of Basic Medical Sciences, School of Medicine, Tsinghua University, Beijing 100084, China. [6]International Institute of Vaccine Research and Innovation (iVac), Savaid Medical School, University of Chinese Academy of Sciences, Beijing 100049, China. [7]These authors contributed equally: Junqing Sun, Lu Kuai, Lei Zhang. ✉E-mail: tianwx@sxau.edu.cn; shiyi@im.ac.cn; dengt@im.ac.cn; gaof@im.ac.cn

a "*trans-acting*" symmetric polymerase dimer to activate its initiation, and an asymmetric dimer, in which one acts as a replicase and the other acts as an encapsidating polymerase, to serve as a replication platform for the viral RNA genome (Carrique et al, 2020; Fan et al, 2019). Once sufficient vRNPs are produced in the nucleus, they will be exported into the cytoplasm in the form of vRNP-M1-NS2 complexes and transported to the plasma membrane for their packaging into progeny virions. Since NS2 has been reported to function in mediating vRNP nuclear export, it has been also named nuclear export protein (NEP) (Martin and Helenius, 1991; O'Neill et al, 1998).

In recent years, the high-resolution structures of the nearly complete FluPols of influenza A, B, C, D viruses have been resolved. The overall structures of FluPols are similar with the central core consisting of PB1, PB2-N, and PA-C surrounded by several flexible peripheral domains including PA-N (containing endonuclease domain) and the C-terminal two-thirds of PB2 (PB2-C) (containing the cap-binding, 627, and NLS domains). Depending on their binding to different structural ligands (e.g., Capped RNA, vRNA/cRNA promoter, the Pol II CTD or host ANP32 family proteins), these peripheral domains can be repacked flexibly around the core region to form different functional states of FluPol (Wandzik et al, 2021; Zhu et al, 2023). The two most representative conformations are the transcriptase conformation and the replicase conformation. The transcriptase conformation enables cap-snatching to occur and its binding to the pS5-Pol II CTD can further stabilize the FluPol transcriptase conformation, thereby enhancing transcription (Lukarska et al, 2017; Serna Martin et al, 2018). The replicase conformation favors dimer formation, including symmetric and asymmetric dimers, to specifically facilitate virus genome replication (Carrique et al, 2020; Fan et al, 2019).

To ensure maximum virus multiplication, a switch from transcription to replication must occur at a later stage in the virus life cycle. It has been proposed that the amount of FluPol and NP, the degradation extent of Pol II, and the production of virus-derived small viral RNAs may play roles in this switch (Perez et al, 2010; Vreede et al, 2010; Vreede et al, 2004). We recently found that high levels of viral NS2 proteins, a spliced product from the NS segment of the virus that is expressed relatively late during infection, can potently inhibit transcription and promote replication (Zhang et al, 2024; Zhang et al, 2023). However, the underlying molecular mechanisms of this regulation were not clear. Moreover, only the crystal structure of the C-terminal domain (CTD) of NS2 has been determined in past years (Akarsu et al, 2003), and the structure of full-length NS2 is yet unsolved, which severely hindered the molecular understanding of the multifunctional roles of NS2 during the influenza virus life.

In this study, we have solved the cryo-EM structure of FluAPol in complex with the viral NS2 protein in which NS2 is dimerized to stabilize the symmetric FluAPol dimers in a highly ordered barrel-like hexamer. Our structural and functional analyses show that the viral NS2 protein facilitates the switch from transcription to replication by affecting FluAPol dissociation from host Pol II and by promoting the transition of FluAPol from its transcriptase conformation into its replicase conformation. Moreover, the structure of the full-length NS2 is also visualized for the first time as part of the NS2-FluAPol complex, which significantly facilitates the molecular understanding of multifunctional roles of NS2 during the influenza virus life cycle.

# Results

## Overall structure of the FluAPol-NS2 complex

We co-expressed the FluPol of influenza A virus (FluAPol) alongside NS2 proteins using the baculovirus expression system and obtained two forms of FluAPol-NS2 complex by gel filtration, a large one and a small one (Fig. EV1). Further SDS-PAGE analysis showed that the intensity of the NS2 protein band is much stronger in the large complex than that in the small complex (Fig. EV1). We then applied cryo-electron microscopy (cryo-EM) to solve the structure of FluAPol in complex with NS2 at a global resolution of ~3.0 Å, using the large protein complex (Fig. EV2; Appendix Table S1).

In this structure, three NS2 domain-swapped dimers arrange three canonical FluAPol 'dimers of heterotrimers' into a symmetric barrel-like 'hexamer of heterotrimers' superstructure, with approximate dimensions of $251 Å \times 247 Å \times 184 Å$ (Fig. 1A,B). The complex formation is achieved by an elongated shape of the helix-bundle-like arrangement of NS2 binding simultaneously to two neighboring FluAPol units. The symmetric FluAPol dimer is organized into a similar conformation as previously seen in both human H3N2 and avian H5N1 FluAPol structures (Appendix Fig. S1), with this symmetric architecture required for vRNA synthesis during replication of the viral genome (Fan et al, 2019). In the structure of each polymerase protomer, the PA and PB1 subunits could be modeled nearly in full-length, except the arch region in the PA subunit, and the β-ribbon, finger tips and priming loop in the PB1 subunit. These elements are involved in promoter binding and RNA synthesis, and could make conformational changes to coordinate different states (Hengrung et al, 2015; Peng et al, 2019; Pflug et al, 2014; Reich et al, 2014). Cryo-EM density for the mid, Cap, 627 domains as well as the Cap-627 linker of PB2 was absent due to conformational flexibility. The Cap-binding domain of PB2 could capture the cap structure of host pre-mRNAs and then rotate in a large degree to align with the endonuclease domain of PA for cap-snatching (Reich et al, 2014). Previous studies of influenza virus polymerases have revealed that the C-terminal domain of PB2 exhibits a higher degree of flexibility in solution without the addition of capped-RNA and vRNA promoters (Fan et al, 2019; Peng et al, 2019), and the similar conformational flexibility of the PB2 C-terminus observed for the FluAPol-NS2 hexamer complex indicated that the polymerase should not be in the transcription state. In the hexamer complex, the three symmetric FluAPol dimers were almost identical and could be well overlaid. In contrast, the three NS2 domain-swapped dimers showed almost the same conformation at one end but a significantly different conformation at the other end, mainly due to the flexible hinge region of NS2, which leads to the varied distance of adjacent FluAPol dimers (Fig. EV3). Thus, the FluAPol-NS2 hexamer lacks standard symmetry and might be dynamic.

## NS2 forms a pseudo two-fold domain-swapped dimer

In the life cycle of influenza virus, NS2 is a multifunctional component which plays critical roles for transporting RNPs and for regulating virus genome transcription and replication, implying potential conformational changes in different states (O'Neill et al, 1998; Robb et al, 2009). Many research groups have tried to

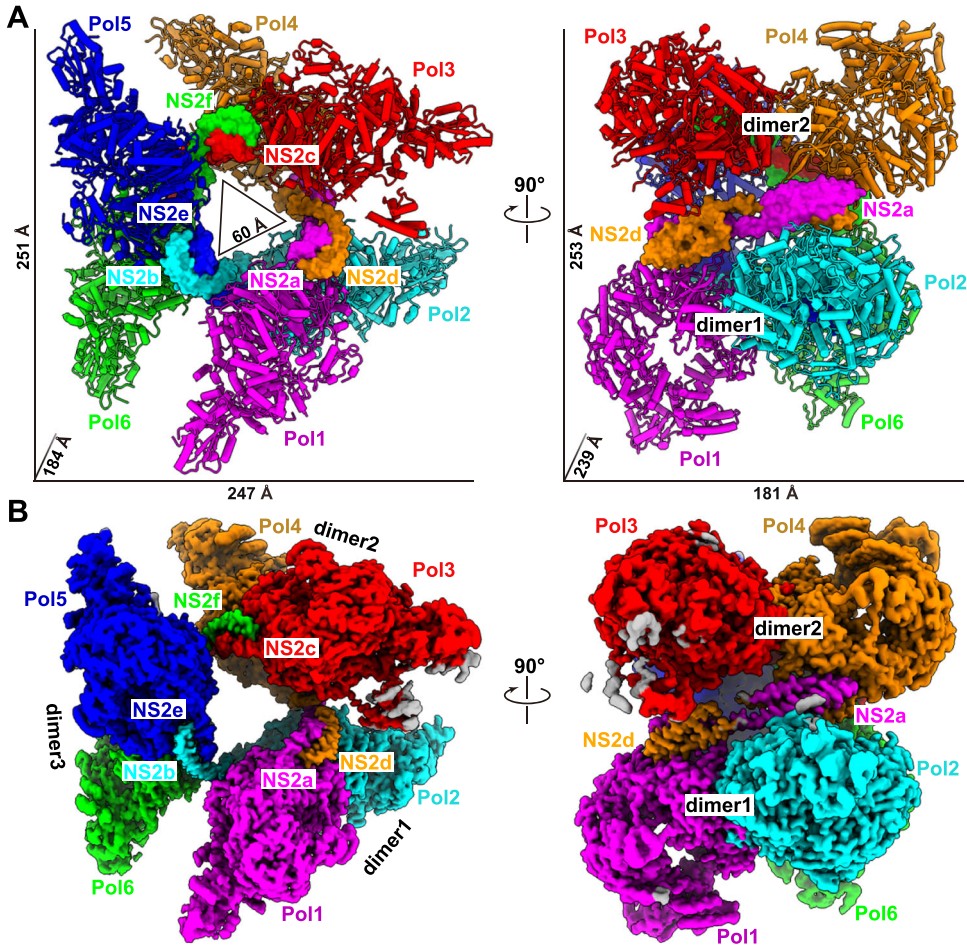

**Figure 1. Overall structure of the influenza A polymerase-NS2 complex.**

Three dimeric influenza A polymerase heterotrimers form a hexamer linked by three NS2 domain-swapped dimers. Each protomer of the polymerase heterotrimers is differently colored and labeled Pol1 to Pol6. The color of the NS2 protomer is the same as the relevant polymerase heterotrimer, which interacts with the N terminus of NS2. (A) The atomic model of influenza A polymerase-NS2 complex, the polymerase and NS2 shown as cartoon and surface, respectively. (B) The composited cryo-EM map of the influenza A polymerase-NS2 complex.

determine the structure and conformation of full-length NS2, however, only a truncated structure of NS2 (residues 54–116) was solved, with the C-terminal domain (CTD) of NS2 forming a dimer through hydrophobic interactions (Akarsu et al, 2003). To solve the structure of full-length NS2, we expressed and purified NS2 using the baculovirus expression system. Full-length NS2 is a monomer in solution, confirmed by size-exclusion chromatography and analytical ultracentrifugation (Appendix Fig. S2). This result was consistent with the previous study, showing that the N-terminus of NS2 prevented the formation of NS2 CTD dimers (Akarsu et al, 2003). We used this purified full-length NS2 for crystallization. Unfortunately, we failed to get any crystals mainly due to the high flexibility of NS2, as in previous trials (Lommer and Luo, 2002; Shtykova et al, 2017).

Interestingly, in the context of the FluAPol-NS2 complex, full-length NS2 formed a pseudo two-fold domain-swapped dimer, which was stabilized by extensive interactions between the N-terminal domain (NTD) of one NS2 protomer and the C-terminal domain (CTD) from the other protomer (Figs. 2A,B

and EV3). Based on the EM density map, we could build almost all the residues of NS2 except for the first four residues. Both the NTD and CTD form a similar helical hairpin structure, and the hinge region between NTD and CTD is relatively flexible with different conformations observed (Figs. 2A,B and EV3). Further structural analysis revealed that hydrophobic interactions and hydrogen bonds are responsible for the formation of the domain-swapped NS2 dimer. It can be seen that intimate hydrophobic interactions are formed by a significant number of residues between the NTD of one NS2 protomer and the CTD of the other NS2 protomer (Fig. 2C). Moreover, a number of hydrogen bonds were also formed between the NS2-NTD and NS2-CTD of the two promoters (Fig. 2C). It is worth noting that the previously determined crystal structure of the NS2 CTD revealed that the NS2 CTD could form a dimer mainly through hydrophobic interactions, and several critical residues are shared in the NS2 domain-swapped dimer (Akarsu et al, 2003) (Fig. EV4).

We performed mutagenesis analysis of the key residues of NS2 that stabilize the NS2 domain-swapped dimer in a cell-based

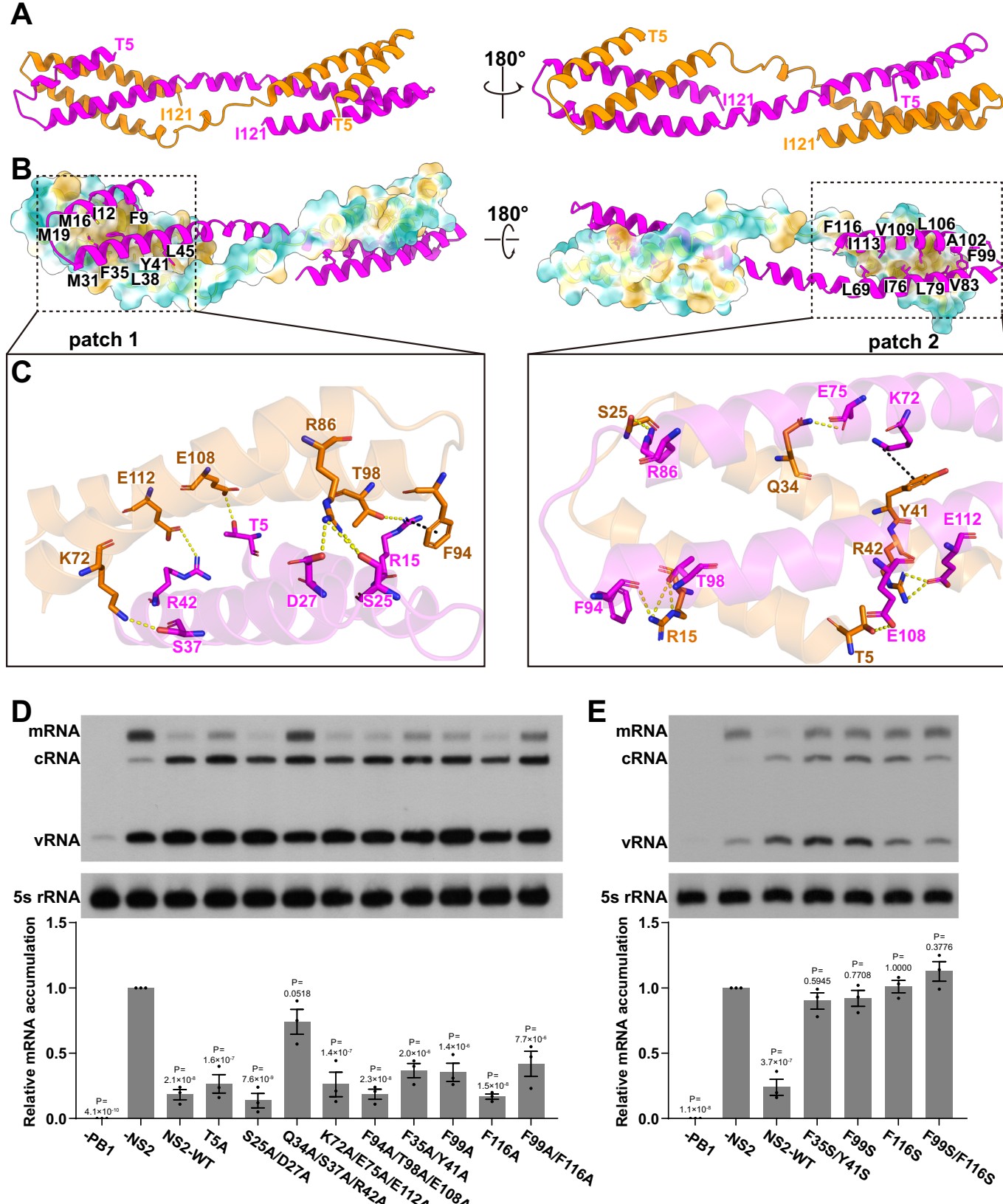

Figure 2.   Structure of NS2.

(A) The domain-swapped dimeric structure of NS2 is shown. The modeled N-terminal residue T5 and the C-terminal residue I121 are indicated. (B, C) There exist extensive interactions between the N-terminal domain of one protomer and the C-terminal region of the other protomer, including hydrophobic interactions (B) and hydrogen bonds (C). (D, E) Effects of different NS2 substitutions on RNP activity in the mini-replicon system. Residues maintaining the dimeric formation of NS2 are substituted to alanine (D) or hydrophilic serine (E). These substitutions will decrease the inhibitory effects of NS2 on influenza A polymerase transcription activity. Data are mean ± s.e.m. from three biological replicates. P values were calculated by one-way ANOVA with Dunnett's post hoc test. Source data are available online for this figure.

mini-replicon system derived from a influenza/WSN/33 virus (Fodor et al, 2002). In this experiment, 293T cells were transfected with NS2 or NS2 mutant expressing plasmids, together with the RNP reconstitution plasmids. The three viral RNA species were simultaneously examined by primer extension analyses at 24 h post-transfection. It is necessary to point out that the regulatory role of NS2 on viral RNA transcription and replication analyzed using this system should not interfere with its essential function as a nuclear export protein, because NS2 can only exert its vRNP nuclear export function by forming vRNP-M1-NS2 complexes (O'Neill et al, 1998; Paterson and Fodor, 2012). To verify this, we further conducted an in situ single-molecule inexpensive FISH (smiFISH) experiment which showed directly that vRNPs were restricted to the nucleus when NS2 was expressed alone, in contrast to substantial cytoplasmic vRNP localization in those cells expressing additional M1 protein (Appendix Fig. S3). As expected, the expression of wild-type NS2 shows significantly reduced mRNA levels but increased cRNA/vRNA levels compared to the control where the empty expression vector was transfected. Interestingly, we found that the triple mutant Q34A/S37A/R42A significantly disrupts the transcription inhibition activity of NS2, while the single mutants T5A, F99A, and the double mutants F35A/Y41A, F99A/F116A, show reduced inhibitory effects (Fig. 2D). Other substitutions, including S25A/D27A, K72A/E75A/E112A, F94A/T98A/E108A, and F116A have no obvious inhibitory effects (Fig. 2D). Moreover, when the hydrophobic residues F35, Y41, F99, and F116 were mutated to a hydrophilic serine, all single or double substitutions completely abolished the inhibitory effects of NS2 on transcription (Fig. 2E). By comparison, all these amino acid substitutions would not visibly change the replication promoting effects of NS2, particularly for vRNA to cRNA synthesis (Fig. 2D,E). Moreover, these critical residues are highly conserved in NS2 of all known influenza A viruses (Appendix Fig. S4).

## Interaction between NS2 and FluAPol

Further inspection of the cryo-EM structure revealed that each NS2 domain-swapped dimer binds to two FluAPol complexes, and the terminal helical hairpins interact with the PA and PB1 subunits of FluAPol (Fig. 3A). Residues K18 and R15 of the N-terminal domain of NS2 form two hydrogen bonds with residue E629 of PA and residue D2 of PB1, respectively (Fig. 3B,C). In addition, residue Q111 of the C-terminal domain of the other NS2 protomer forms a hydrogen bond with residue S558 of PA (Fig. 3B,C). If we superimposed the polymerase protomer (Pol1) onto the previously determined crystal structure of apo H3N2 polymerase, the NS2 binding has little influence on the conformation of polymerase except that the flexible loop between residues 550–556 of PA subunit could now be modeled and stabilized by the interaction with NS2. At the same time, the β-sheet formed from residues 618–632 of the PA subunit is slightly adjusted which allows its

interaction with K18 of NS2 (Fig. EV5). However, if we took the polymerase dimer as a unit, we found that the binding of NS2 made the polymerase dimer less compact and the other protomer is offset by ~5 Å (Fig. EV5).

We further validated the key residues responsible for the NS2-FluAPol interaction using the cell-based mini-replicon system. As expected, the single amino acid substitution R15A can destroy the transcription inhibition activity of NS2, while K18A substitution reduces the inhibition activity. Q111A substitution has no obvious effect on the inhibition activity. In addition, the R15A/K18A double substitution and R15A/K18A/Q111A triple substitution can also destroy the transcription inhibition activity of NS2 (Fig. 3D,E).

## NS2 hampers the interaction between FluAPol and the Pol II CTD

Since the transcriptional activity of FluAPol absolutely relies on its interaction with the CTD of human RNA polymerase II (Pol II) to enable cap snatching, we superimposed our NS2-FluAPol complex structure onto the structure of H17N10 FluPol in complex with the CTD of Pol II. Interestingly, we found that the NS2 interaction site overlaps with the binding site of CTD of Pol II (Fig. 4A–C). It indicates that NS2 binding will sterically hamper the interaction between FluAPol and the Pol II CTD, thereby inhibiting genome transcription processes.

To validate the effect NS2 has on interfering with FluAPol association with the Pol II CTD, we further developed a cell-based NanoBiT luciferase complementation reporter assay which allows for quantitative investigation of the interaction between FluAPol and the Pol II CTD in the presence of wild-type NS2 or mutant NS2 (Fig. 4D; Appendix Fig. S5). Indeed, it clearly showed that the co-expression of wild-type NS2 protein could significantly reduce the interaction between FluAPol and the Pol II CTD, while the representative transcription-inhibition disrupting NS2 mutants (R15A/K18A/Q111A, F35S/Y41S, and F99S/F116S) lose this effect (Fig. 4E). These results strongly suggest that NS2 could sterically hamper the interaction between FluA Pol and the Pol II CTD.

## NS2 prevents FluAPol from adopting the transcriptase conformation

The structures of FluAPol in different functional states have been resolved. In the transcriptase conformation, the cap binding domain of PB2 and endonuclease domain of PA-N should be aligned for cap-snatching process in which the 627 domain of PB2-C locates close to the PA-C, and distant from the PA-N. By contrast, in the replicase conformation, the PB2-C domain is flipped up so that the 627 domain is distant from PA-C and enables the NLS domain of PB2-C to bind to the in situ rotated PA-N. Interestingly, a superimposition of our NS2-FluAPol complex structure onto the structure of H3N2 FluAPol in the transcriptase conformation shows that the bound NS2 will also have a steric

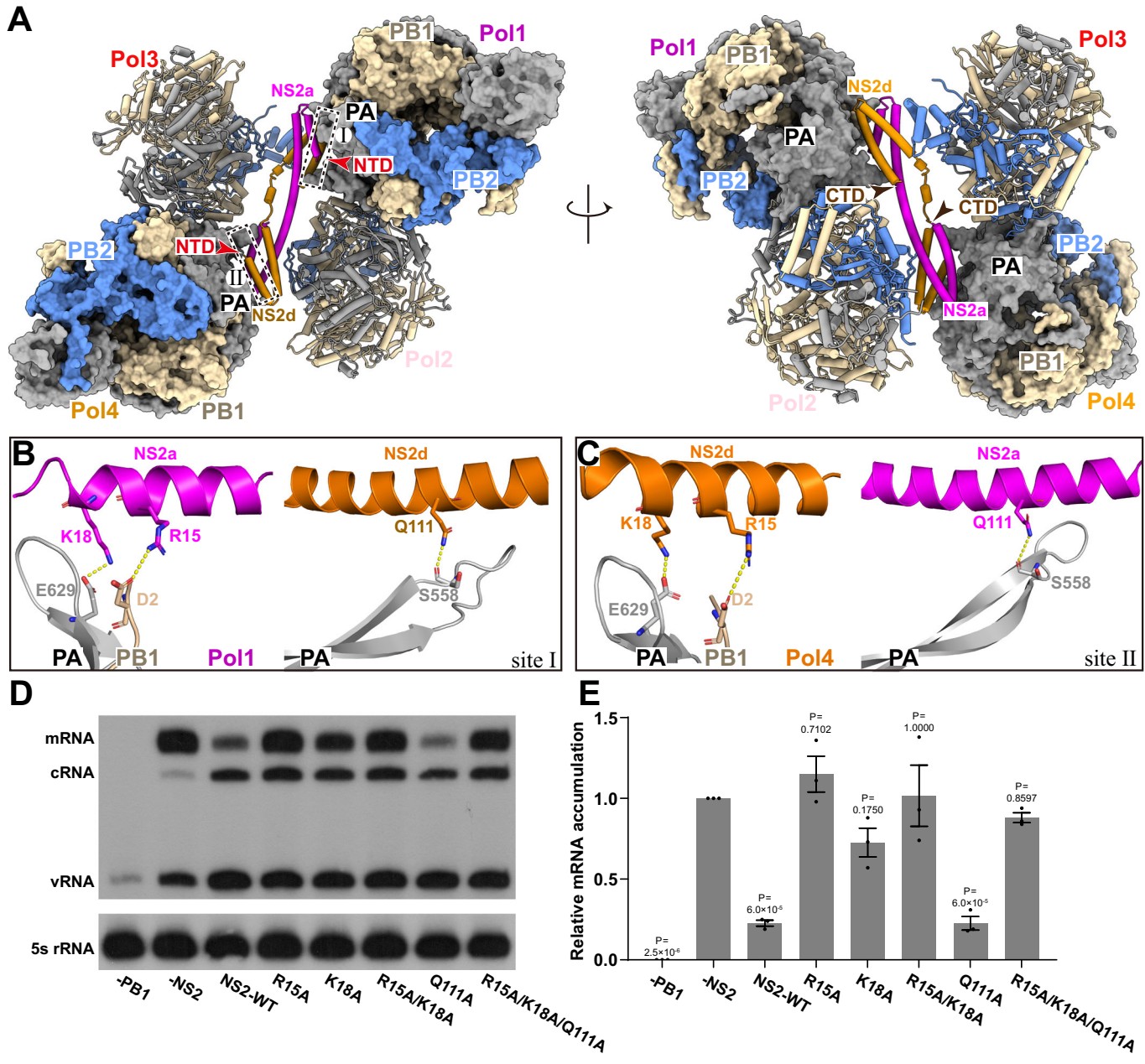

**Figure 3. Interaction between FluAPol and NS2.**

(A) Interface of FluAPol-NS2 complex. A domain-swapped NS2 dimer and two FluAPol symmetric dimer are shown to display the interaction site. (B, C) Detailed interaction between FluAPol and NS2. The hydrogen bonds are shown as yellow dashed lines. (D) Inhibitory effects of NS2 substitutions on RNP transcription activity. (E) Quantification of the inhibitory effects of NS2 substitutions on the polymerase transcription activities shown in (D). Data are mean ± s.e.m. from three biological replicates. *P* values were calculated by one-way ANOVA with Dunnett's post hoc test. Source data are available online for this figure.

conflict with the 627 domain of PB2-C in the transcription state (Fig. 5A). Therefore, binding of NS2 prevents FluAPol from adopting the transcriptase conformation.

To functionally explore whether NS2 can promote replication independently of its transcription inhibition activity, we examined the effect of NS2 on promoting viral RNA replication only by using a transcription-defective FluA Pol mutant (PA D108A) in the cell-base mini-replication system. The PA D108A point mutation has been reported to be able to abolish endonuclease activity of PA and thus

deprive FluA Pol of its cap-snatching capability that is required for transcription initiation (Hara et al, 2006; Yuan et al, 2009). The results clearly showed that NS2 can promote replication in a dose-dependent manner when transcription is completely inhibited in the system (Fig. 5B,C), suggesting that the effect of NS2 on promoting replication is independent of its effect on inhibiting transcription.

Taken together, based on our structural analyses and functional studies, we conclude that binding of NS2 to FluAPol not only sterically hampers the interaction between FluAPol and the Pol II

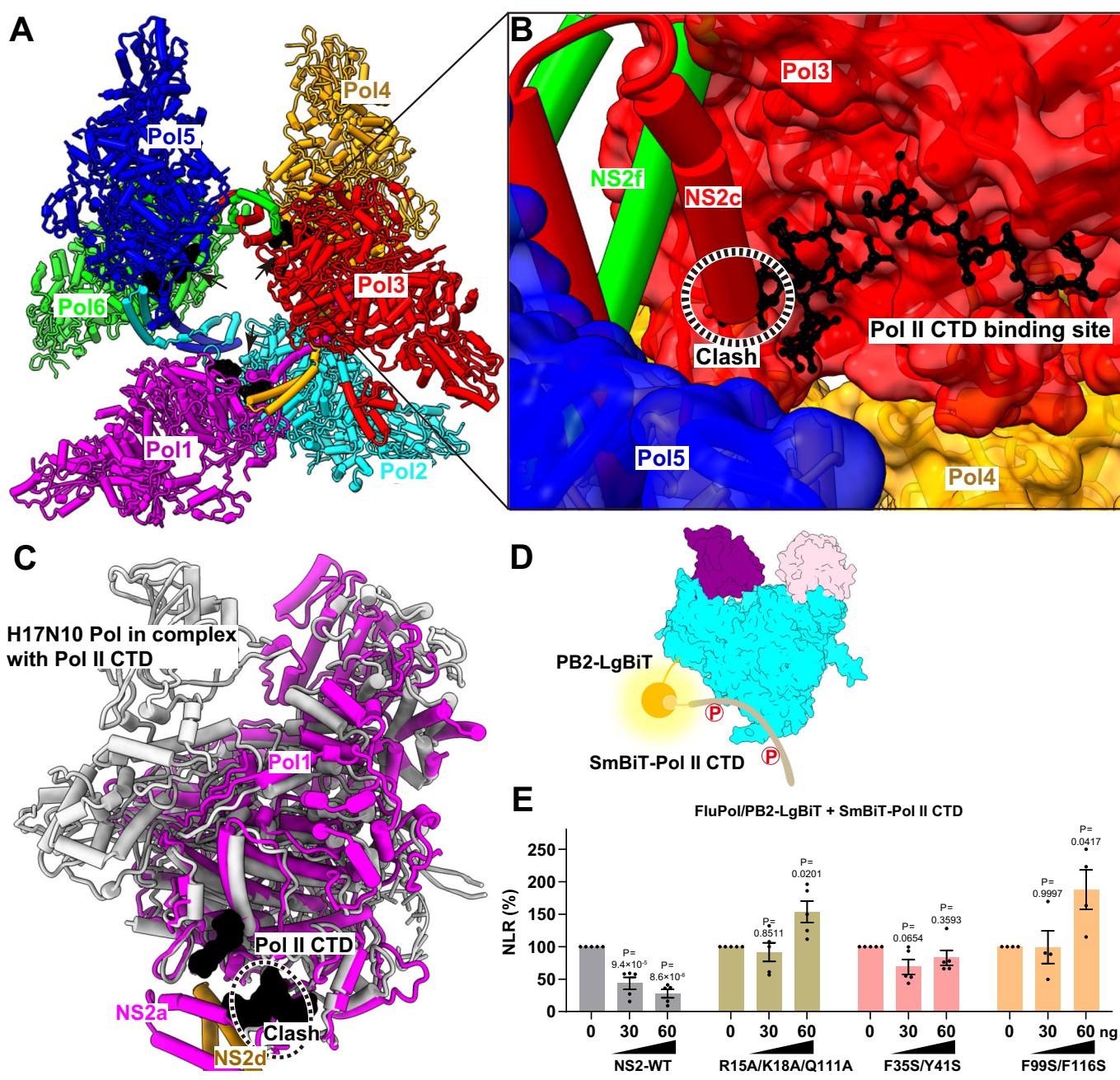

Figure 4. NS2 hampers the interaction between FluAPol and the Pol II CTD.

(A) The model of FluAPol-NS2 hexamer bound to the C-terminal domain of human RNA polymerase II (Pol II CTD). The Pol II was modeled based on the structure of H17N10 polymerase-Pol II CTD complex (PDB ID:5M3H). The FluAPol-NS2 hexamer is shown as cartoon and colored as in Fig. 1. The Pol II CTD is shown as solvent accessible surface and colored in black. The Pol II CTD binding sites of influenza A polymerases are inaccessible in the hexamer, as these sites are oriented towards the interior of the hexamer. (B) Close-up view of Pol II binding sites in the influenza A polymerase-NS2 hexamer. The Pol II CTD is shown as ball and stick and colored in black. The NS2c and NS2e are shown as cartoon. The region with steric conflicts is indicated by a dashed oval. (C) Superimposition of the H17N10 polymerase-Pol II CTD complex and FluAPol-NS2 complex structures. The Pol II CTD is presented in surface and colored in black. The binding sites of the Pol II CTD and NS2 overlapp, suggesting that the NS2 protein sterically hinders the interaction between FluAPol and the Pol II CTD. (D) Schematic of the NanoBiT luciferase complementation-based assay to analyze the interaction between FluPol and the Pol II CTD. The large BiT subunit (LgBiT, fused to the C-terminus of PB2, PB1, or PA of FluPol) and the small BiT subunit (SmBiT, fused at the N-terminus of the Pol II CTD) interact to reconstitute an active Nano luciferase enzyme when FluPol and the Pol II CTD are associated. (E) Effects of wild-type NS2 or NS2 mutants on the interaction between FluAPol and the Pol II CTD. The results are shown as normalized luminescence ratio (NLR) and are as mean ± s.e.m. of four or five biological replicates. P values were calculated by one-way ANOVA with Dunnett's post hoc test. Source data are available online for this figure.

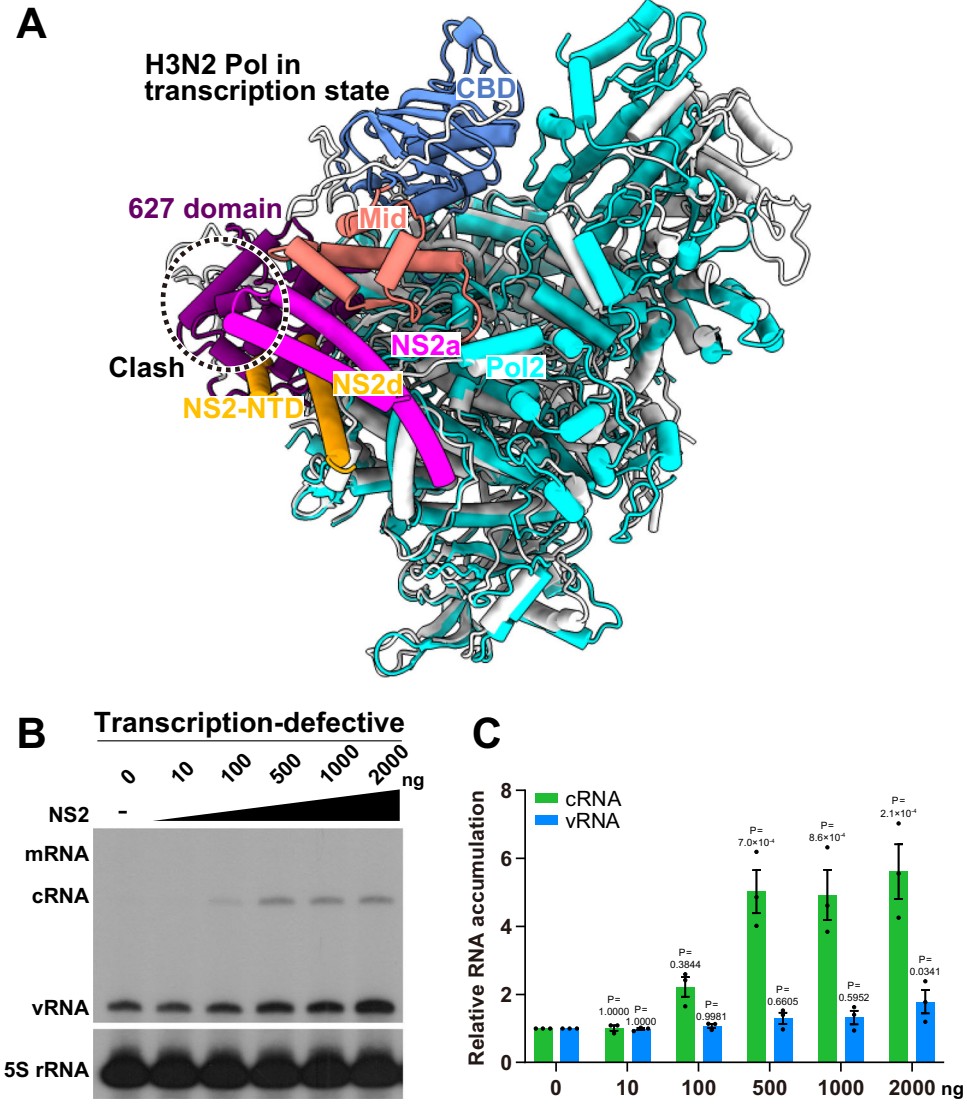

**Figure 5. NS2 prevents FluAPol from adopting the transcriptase conformation.**

(A) Structural comparison of H3N2 polymerase in the transcription state (PDB ID:6RR7) and a FluAPol protomer from the FluAPol-NS2 hexamer complex. During transcription, the cap binding domain of PB2 and the endonuclease domain of PA should be aligned to enable the cap-snatching process. In the structure of the FluAPol-NS2 complex, binding of NS2 would cause a steric clash with the 627 domain of PB2 and thus prevents FluAPol from adopting the transcriptase conformation. (B) Dose-dependent effect of NS2 on the accumulation of viral RNAs in the transcription-defective (PA-D108A) RNP reconstitution system in HEK 293T cells. (C) Quantification of the effects of NS2 on viral RNA synthesis shown in (B). The graph shows the relative mean intensities of viral cRNA and vRNA normalized to 5S rRNA. The graph shows the mean ± s.e.m. from three biological replicates. Source data are available online for this figure.

CTD, but also prevents FluAPol from adopting the transcriptase conformation. These structural findings are consistent with the above functional observations that high levels of NS2 can not only inhibit viral mRNA synthesis, but also independently promote c/vRNA synthesis.

## Discussion

In the past years, we have gained substantial structural and functional insights into how the influenza virus polymerase replicates and transcribes the viral genome (Wandzik et al, 2020;

Zhu et al, 2023). The influenza virus polymerase can adopt different conformations to achieve different functional states to complete viral RNA transcription and replication. Moreover, the asymmetric or symmetric forms of influenza polymerase complexes, and the participation of host ANP32 family proteins and RNA polymerase II, are necessary for efficient genome replication and transcription (Carrique et al, 2020; Wang et al, 2022). Previous studies have revealed that the profiles of mRNA, cRNA, and vRNA accumulation during infection are delicately and dynamically regulated, exhibiting a notable and significant switch from genome transcription to replication occurring in the later stage of an infection (Lukarska et al, 2017). We recently reported that the non-structural

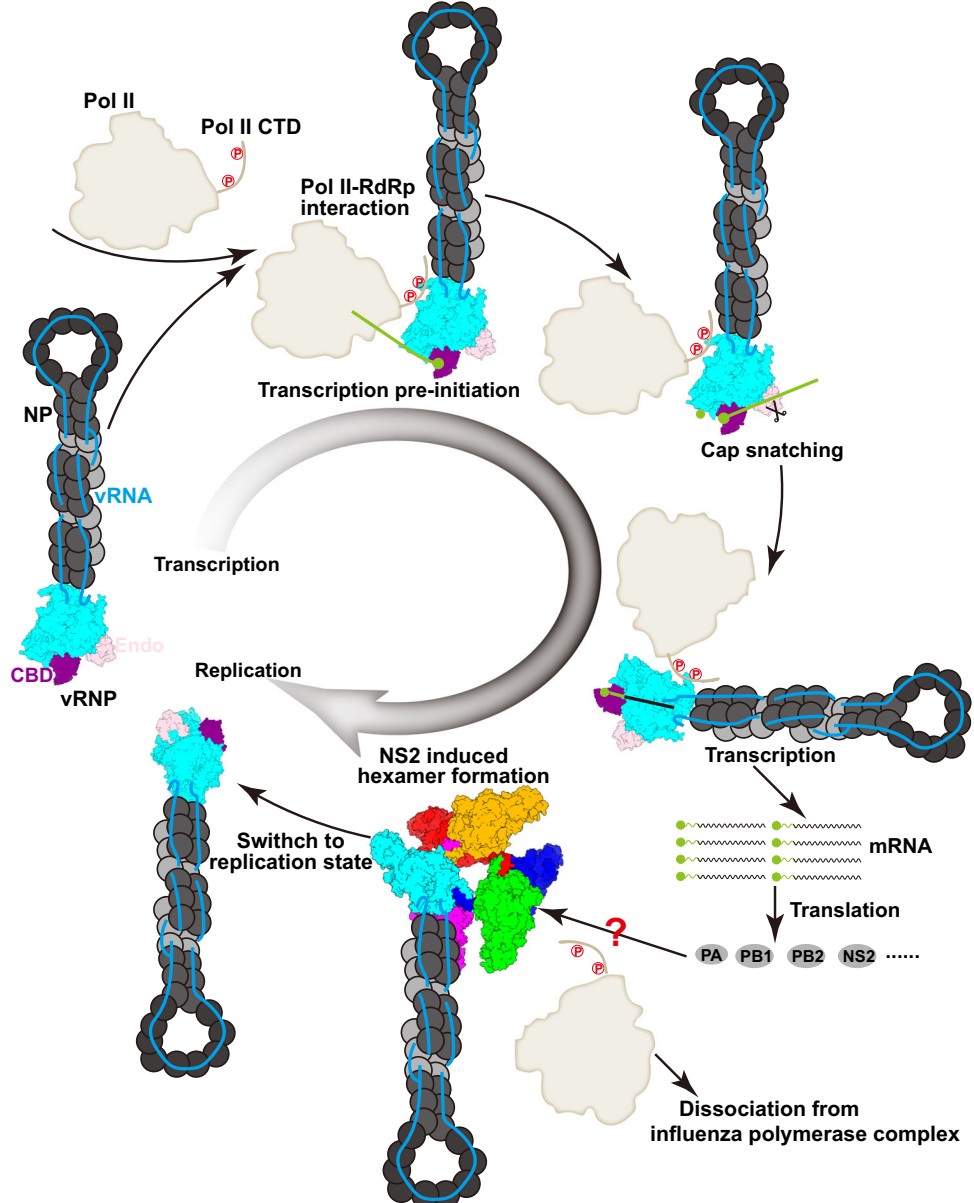

**Figure 6. Model of FluAPol switching from transcription to replication states.**

After virus entry, vRNPs are released and transported into nucleus. To initiate genome replication, the FluAPol would form asymmetric or symmetric dimers with the newly synthesized polymerase. Thus, at early stages, the major Pol II activity is genome transcription that depends on the binding to the Pol II CTD, which facilitates the cap-snatching process. The influenza polymerase core is colored in cyan, the cap binding domain (CBD) in purple, the endonuclease domain (Endo) in pink and Pol II in white. The mRNA transcripts are then translated into viral proteins in the cytoplasm, and then the PA-PB1, PB2, and NS2 are transported to the nucleus to regulate polymerase activity. The domain-swapped NS2 dimer binds to the polymerase and induces the formation of FluAPol-NS2 hexamers to inhibit the transcription process.

proteins (NS1 and NS2) of influenza viruses are involved in fine-tuning the dynamic syntheses of the three viral RNA species (Zhang et al, 2023). We found that NS2 can inhibit transcription and promote replication through its N-terminal amino acids 1–20 and the last C-terminal residue 121, respectively (Zhang et al, 2023). Here, we show the molecular basis for regulation of influenza virus genome transcription and replication by the NS2 protein, and that NS2 binding dissociates the host RNA polymerase II from the polymerase because of steric conflicts and prevents the polymerase from adopting the transcriptase conformation. We thus

propose that NS2 functions as a molecular transfer-switch, inhibiting transcription but promoting replication. At early stage of infection, when transcription is cued, and when host RNA polymerase II and NS1 protein are widely available, the influenza polymerase is more prone to adopt the transcriptase conformation (Fig. 6). As infection progresses, replication promoting cues, with the binding of newly produced viral polymerase and spliced NS2 protein forming a barrel-like hexamer, thereby decreasing the association of viral polymerase with host RNA polymerase II and inhibiting viral RNA transcription (Fig. 6).

Previous studies showed that the influenza virus NS2 protein could exist as both a monomer and a dimer in solution, depending on the solution components (Shtykova et al, 2017). Here, we captured the domain-swapped dimer form of the NS2 protein that interacts with the symmetrically-dimeric FluAPol. Our mutagenesis work with the mini-replicon system and the cell-based NanoBiT complementary reporter assay revealed that the substitutions of key residues responsible for the NS2-FluAPol interaction could hamper the transcription inhibitory effect of NS2 protein by interfering with the interaction between FluAPol and the Pol II CTD, however, the replication-related cRNA and vRNA levels were still elevated by both the wild type and mutant NS2 proteins. We further confirmed that the effect of NS2 on promoting viral RNA replication is independent of its effects on inhibiting viral RNA transcription in a transcription-defective mini-replicon system. All these facts indicate that there should exist other interaction modes between FluAPol and NS2, in addition to the FluAPol-NS2 hexamer observed here. We previously showed that the last residue (I121) of NS2 is crucial for the function of NS2 in promoting replication, and we deduced that the CTD of NS2 might be a participant in the replication-promoting interaction mode (Zhang et al, 2024; Zhang et al, 2023). In addition, previous studies have shown that the NS2 CTD could form a helical hairpin structure and that the NS2 NTD was predicted to be disordered (Akarsu et al, 2003). Our complex structure revealed that full-length NS2 forms a domain-swapped dimer which is stabilized by interactions between the NTD and the CTD of different NS2 protomers, with the NTD also forming a helical hairpin structure. However, we cannot rule out the possibility that the NS2 NTD might be flexible and adopt other conformations, which should be further studied in the future. Nevertheless, the NS2-FluAPol complex structure resolved here offers the first view of the full-length NS2 which may significantly facilitate elucidating molecular mechanisms of other important functions played by NS2 during influenza infection.

The NS2 protein of influenza virus was renamed as NEP (nuclear export protein) by O'Neill et al, in 1998 because they mapped a HIV Rev-like nuclear export signal (NES1) in NS2 (O'Neill et al, 1998). It is now generally believed that NS2 acts as a protein adapter molecule in mediating vRNP nuclear export through a Crm1-dependent nuclear export pathway by forming vRNP-M1-NEP-Crm1 complex (Neumann et al, 2000; Paterson and Fodor, 2012). Indeed, we found that the expression of NS2 alone with vRNP is incapable of transporting vRNP into the cytoplasm (Appendix Fig. S3). Interestingly, the NES1 identified in NS2 was mapped to 12–21 amino acids in its N-terminus in which a few hydrophobic residues I12, M16, M19, and L21 have been shown to be critical for its NES function, while, in this report, we show R15 and K18 within the NES1 are key residues in mediating NS2-FluPol interactions and play a significant role in inhibiting transcription. This evidence further corroborates that the NS2 NTD is highly flexible and may adopt multiple conformations in fulfilling its functions in regulating viral RNA syntheses and vRNP nuclear export.

In fact, in addition to influenza virus, dynamic regulation of viral transcription and replication by virus-encoded proteins has been previously observed in other negative-sense RNA viruses, including vesicular stomatitis virus, Ebola virus, arenavirus, and respiratory syncytial virus (Asenjo and Villanueva, 2016; Biedenkopf et al, 2016; Hwang et al, 1999). Thus, the self-regulation of viral RNA transcription and replication might be a general mechanism for negative-sense RNA viruses. Further studies of the molecular mechanisms of self-regulation by viral/host factors would greatly facilitate both the development of novel small-molecule inhibitors that disrupt the interaction interface, as well as the design of attenuated live virus vaccines.

# Methods

## Cell lines

Human embryonic kidney HEK 293T (ATCC, CRL-3216) cells were purchased from the American Type Culture Collection (ATCC) and were maintained in Dulbecco's modified Eagle's medium (DMEM; Gibco) supplemented with 10% fetal bovine serum (FBS; Gibco) and 1% penicillin-streptomycin. Cells were cultured in humidified incubators at 37 °C with 5% $CO_2$.

## Protein expression and purification

The three subunits of human influenza A/NT/60/1968 (H3N2) virus polymerase and NS2 were co-expressed in High Five™ cells from codon-optimized genes cloned into pACEBac1 plasmid. The Gibco™ Sf9 cells were used for transfection and virus amplification. The High Five cells were used for protein expression and harvested at 48 h after virus infection. The High Five cells were collected by centrifuge at 4000 rpm for 10 min. We discarded the supernatant, and then resuspended the cells with buffer A containing 25 mM HEPES, 500 mM NaCl, 10% glycerol, 1 mM TCEP, pH 7.5, supplemented with 1 mM PMSF. The cells were disrupted by sonication for 40 min. Cell debris were removed by centrifuge at 12,000 rpm for half an hour. Ammonium sulfate (0.5 g/mL) was added to the supernatant on the magnetic stirrer to mix well. The resulting precipitate was collected using centrifugation (12,000 rpm, 1.5 h) and dissolved with buffer A. After a final ultracentrifuge (30,000 rpm, 1.5 h), the supernatant was filtered with 0.22 μm filter and then flowed through the StrepTrap HP column (Cytiva) for affinity chromatography. The samples were pooled and injected into a Superose™ 6 Increase 10/300 GL column (Cytiva) running in 25 mM HEPES, 500 mM NaCl, 5% glycerol, 1 mM TCEP, 1 mM Desthiobiotin, pH 7.5.

The purification of full-length NS2 was similar to above description. Briefly, the cleared supernatant was loaded onto the HisTrap HP column and then eluted using imidazole gradient. The targeted gradient (~300 mM) was pooled and injected into a Superdex™ 200 Increase 10/300 GL column (Cytiva) running in 25 mM HEPES, 500 mM NaCl, 5% glycerol, pH 7.5. The fractions of protein complex were identified by SDS-PAGE and then concentrated.

## Analytical ultracentrifugation analysis

The purified full-length NS2 was centrifuged at 31,000 rpm and 20 °C using Beckman XLI analytical ultracentrifuge equipped with an An50-Ti rotor in quartz cells containing standard double Sector centerpieces. The absorbance data were collected at 230 and 280 nm over 20 h, with scans recorded every 3 min. Sedimentation velocity data were analyzed using the c(s) distribution method in the software SEDFIT (v15.1).

## Cryo-EM sample preparation and data acquisition

To prepare the cryo-EM sample, the FluPolA-NS2 protein complex solution (4.0 µL, 0.5 mg /mL) was loaded to the holey film grid (Ni-Ti R1.2/1.3, 300 mesh) which was glow discharged for 60 s at 15 mA. Grids were then blotted with blot time 3 s and blot force −5 at a temperature of 4 °C and a humidity level of >99% and plunged into liquid ethane using a Vitrobot Mark IV device (ThermoFisher Scientific). The prepared grids were transferred to a Titan Krios transmission electron microscope (ThermoFisher Scientific, 300 kV) equipped with Gatan K3 detector and GIF Quantum energy filter. Movies were collected at 105,000× magnification with a calibrated pixel size of 0.69 Å over a defocus range of −1.0 µm to −2.0 µm in super resolution counting mode, with a total dose of 60 e⁻/Å² using automated acquisition software EPU (ThermoFisher Scientific).

## Cryo-EM image processing

The detailed data processing workflow is summarized in Fig. EV2. All the raw dose-fractionated images stacks were 2× binned, aligned, dose-weighted, and summed using MotionCor2 (Zheng et al, 2017). The contrast transfer function (CTF) estimation, particle picking and extraction, 2D classification, ab initio model generation, 3D refinements were performed in cryoSPARC v.3.3.1 (Punjani et al, 2017).

For the FluAPol-NS2 complex, a total of 10,967 micrographs were collected. We picked out ~68,000 particles using blob-pick procedure of cryoSPARC from 1000 micrographs, and then these particles were subjected to 2D classification. After three rounds of 2D classification, we selected good particles in different views for Topaz training and then generated the Topaz model. Then we applied the Topaz (Bepler et al, 2019) procedure to select particles against entire micrographs. A total of 2,427,238 initial particles were picked and extracted from 10,967 micrographs. After the extensive 2D classification, approximately 823,434 good particles were selected to generate the initial models for heterogeneous refinement. Among six classes, the fourth class showed the highest integrity and was selected for non-uniform refinement which yielded a density map at 3.11 Å. Then, we performed another round of 3D classification without alignment and obtained three distinct volumes. The two dominant classes which represent 32% and 37% of total particles were identified, and displayed clear features of secondary structural elements. These particles were subjected to non-uniform and CTF refinements in cryoSPARC v.3.3.1, and finally yielded a final density map at 2.89 Å resolution estimated by the gold-standard Fourier shell correlation at cut-off value of 0.143. The final map was sharpened by DeepEMhancer (Sanchez-Garcia et al, 2021). We also conducted one round of local refinement by focusing on each dimeric polymerase and the NS2 domain-swapped dimer to improve the map quality, and finally yielded the 2.97 Å, 3.01 Å, and 3.12 Å maps, which were used to generate a composite map. The densities of NS2 were with highest quality in the 2.97 Å EM density map, thus we analyzed the interactions between NS2 dimer based on this map.

## Model building and refinement

The structure of apo H3N2 polymerase heterotrimer (PDB ID: 6QNW) and C-terminal of NS2 (PDB ID:1PD3) were rigidly docked into the EM density map using Chimera (Pettersen et al, 2004). The N-terminal domain of NS2 was built manually in COOT (Emsley and Cowtan, 2004). The initial coordinates were refined against EM density map in real space using PHENIX (Adams et al, 2010), in which the secondary structure restraints and Ramachandran restraints were applied. The stereochemical quality of each model was assessed using MolProbity (Chen et al, 2010). Structural figures were prepared by Pymol (https://pymol.org/) and CHIMERAX (Goddard et al, 2018).

## Mini-replicon system and primer extension analysis

Approximately 10⁶ HEK 293T cells were co-transfected with 0.5 µg each of the pcDNA plasmids expressing PB2, PB1, PA/PA D108A, NP and 0.5 µg of the NA vRNA expression plasmids (pPOLI-NA) using Lipofectamine 2000 and Opti-MEM according to the manufacturer's instructions. Where required, 1 µg of the pcDNA-TAP-NS2 (either wild type or mutant) was co-transfected. In the mini-replicon system, certain concentration of the empty vector was co-transfected to maintain equal concentrations of the total plasmid transfected. The total RNA was extracted 24 h after transfection using TRIzol reagent (Invitrogen). The steady-state levels of three NA RNA species (mRNA, cRNA, and vRNA) were analyzed by primer extension analysis with ³²P-labeled specific primers for the positive (5′-TGGACTAGTGGGAGCATCAT-3′) and negative (5′-TCCAGTATGGTTTTGATTTCCG-3′) sense NA RNA. The primer (5′-TCCCAGGCGGTCTCCCATCC-3′) was used for the detection of the internal control 5S rRNA. Primer extension products were analyzed on a 6% PAGE gel containing 7 M urea and detected by autoradiography.

## NanoBiT luciferase complementation assay

To detect the interaction between FluAPol and the C-terminal domain of host polymerase II (Pol II CTD), we utilized the SmBiT-tagged Pol II CTD expression plasmid (pCAGGS-SmBiT-Pol II CTD) with the LgBiT-tagged FluPol subunit alone (pCAGGS-PB2-LgBiT, pCAGGS-PB1-LgBiT or pCAGGS-PA-LgBiT) or combined with the other two non-tagged subunits, respectively. The SmBiT-Pol II CTD (S5A), in which all serine 5 residues were replaced with an alanine, was used as negative control. All of the above plasmids were co-transfected in HEK 293T cells for 30 ng. To test the effect of NS2 on the interaction between FluAPol and the Pol II CTD, 30 or 60 ng of pCAGGS plasmids expressing NS2-WT or mutants were co-transfected. For all assays, a certain concentration of the empty vector was co-transfected to maintain equal concentrations of the total plasmid transfected. After 24 h of incubation at 37 °C, the Nano luciferase enzymatic activities were measured using the Nano-Glo Live Cell Assay System (Promega) and a GLOMAX 96 Microplate luminometer (Promega).

## RNA fluorescence in situ hybridization (RNA-FISH)

HEK 293T cells were seeded into 24-well culture plates containing sterile, round coverslips at a final density of 8.3 × 10⁴ cells per well, then co-transfected with indicated plasmids. After 42 h of incubation, cells were fixed with 2% PFA for 30 min, then washed three times with equal volumes of RNase-free Phosphate Buffered Saline. The fixed cells were permeabilized with TE-TritonX100 solution (0.5% TritonX-100,

50 mM Tris-HCl, pH = 8.0, 5 mM EDTA in DEPC treated water) for 15 min. The coverslips were then washed once with 2× SSC buffer and kept in a FISH wash solution containing 30% deionized formamide (Mei5bio), 4 mM Ribonucleoside Vanadyl Complex (NEB), and 2× SSC prior to probe hybridization.

RNA FISH was performed according to published protocols with modifications (Tsanov et al, 2016). For each of the two RNA targets (M1 mRNA and NA vRNA), we designed 23 primary probes with their 5′ 24–26 nt binding to the target RNAs and their 3′ regions hybridizing to the cognate fluorescently labeled detection (secondary) probes (5′-Cy3-TGCCGTGGATCAAGTGCCGACC-Cy3-3′ for M1 mRNA; 5′-Cy5-CGGCCTCGTACCACAATGCGGA-Cy5-3′ for NA vRNA). Pooled primary probe set and fluorescently labeled secondary probes were mixed at a final concentration of 5 µM and 10 µM, respectively, and annealed by incubating the probe mixtures at 95 °C for 5 min then cooling down to 12°C. RNA-FISH hybridization mix were prepared by combining 1× volume of the annealed probe mixtures with 24× volumes of FISH hybridization solution (30% formamide, 0.5 mg/ml yeast tRNA, 2× SSC, 4 mM vanadyl-ribonucleoside complex, and 10% dextran sulfate in DEPC treated water). The coverslips were immersed in 80 µL aliquots of RNA-FISH hybridization mix, sealed with parafilm and incubated overnight at 37 °C. After overnight incubation, the coverslips were washed twice using FISH wash solution with a 15 min, 37 °C incubation period preceding buffer exchanges. The cells were then washed once with RNase-free PBS and immersed in PBS containing 5 µg/mL DAPI for 15 min at room temperature. The coverslips were washed with DEPC-treated double-distilled water and mounted on a glass slide using the anti-bleaching mounting solution Fluoromount (Yeasen Biotech.) and sealed with colorless nail polish.

### Confocal microscopy and quantitative image analysis

Multicolor z-stack images were acquired on a Leica confocal SP8 microscope using a high-NA 100× oil objective. Image analysis were performed with Fiji (Schindelin et al, 2012), the Python package Scikit-Image (van der Walt et al, 2014), and bespoke Python scripts. In brief, single-cell outlines were obtained by manual annotation using the Fiji polygon ROI tool. Nucleus segmentation was achieved by applying a watershed segmentation algorithm on DAPI fluorescence images. Binary masks of the filled cell polygon and the nucleus were then used to define the cytosolic (cell mask devoid of the nucleus mask) and the nucleus fractions of each cell. NA positive cells were defined as the cells whose 90th percentile cellular Cy5 fluorescence intensity exceeded 20 a.u. (about 4 times over cellular average). Image crops were rendered using customized Python scripts powered by microfilm (guiwitz.-github.io/microfilm).

## Data availability

The cryo-EM density map and atomic coordinate have been deposited to the Electron Microscopy Data Bank (EMDB, www.emdataresource.org/) and the Protein Data Bank (PDB, www.rcsb.org), respectively. The composite hexameric structure of FluAPol-NS2 complex (EMD-39022, www.ebi.ac.uk/emdb/EMD-39022; PDB ID: 8Y7O, https://doi.org/10.2210/pdb8Y7O/pdb). The local dimeric structure of FluAPol-NS2 complex

(EMD-39020, www.ebi.ac.uk/emdb/EMD-39020; PDB ID: 8Y7M, https://doi.org/10.2210/pdb8Y7M/pdb).

The source data of this paper are collected in the following database record: biostudies:S-SCDT-10_1038-S44319-024-00208-4.

## Peer review information

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

## Acknowledgements

We thank all staff at the cryo-EM Center, Shanxi Academy of Advanced Research and Innovation for their technical supports on the cryo-EM data collection. The study was supported by the National Key Research and Development Program of China (2022YFF1203200 to TD and 2021YFC2300700 to YS and TD), Strategic Priority Research Program of CAS (XDB29010000 to GFG and YS), National Natural Science Foundation of China (NSFC) (81871658, 32192452, 32100119, 31870160, and 32070173 to YS, QP, YL, and TD), Shanxi Key R&D Program (201102130501001 to WXT), The Earmarked Fund for Shanxi Agriculture Research System (2023-07, 2024CYJSTX15 to WXT) and The Special Fund for Science and Technology Innovation Teams of Shanxi Province (202204051001022 to WXT) and Beijing Natural Science Foundation (M22031 to TD).

## Author contributions

**Junqing Sun**: Data curation; Software; Validation; Visualization; Writing—original draft; Writing—review and editing. **Lu Kuai**: Resources; Data curation; Validation. **Lei Zhang**: Conceptualization; Validation; Writing—original draft; Writing—review and editing. **Yufeng Xie**: Conceptualization; Resources; Data curation; Software; Formal analysis; Supervision; Methodology. **Yanfang Zhang**: Resources; Software; Methodology. **Yan Li**: Conceptualization; Resources; Supervision. **Qi Peng**: Conceptualization; Resources; Supervision; Writing—original draft; Writing—review and editing. **Yuekun Shao**: Data curation; Software. **Qiuxian Yang**: Resources; Software. **Wen-xia Tian**: Supervision; Methodology. **Junhao Zhu**: Validation; Methodology. **Jianxun Qi**: Conceptualization; Resources; Data curation; Formal analysis; Supervision; Funding acquisition; Validation; Visualization; Writing—original draft; Project administration; Writing—review and editing. **Yi Shi**: Conceptualization; Resources; Supervision; Funding acquisition; Validation; Methodology; Writing—original draft; Project administration; Writing—review and editing. **Tao Deng**: Conceptualization; Formal analysis; Supervision; Funding acquisition; Investigation; Visualization; Writing—original draft; Project administration; Writing—review and editing. **George F Gao**: Conceptualization; Resources; Formal analysis; Supervision; Funding acquisition; Validation; Methodology; Writing—original draft; Project administration; Writing—review and editing.

Source data underlying figure panels in this paper may have individual authorship assigned. Where available, figure panel/source data authorship is listed in the following database record: biostudies:S-SCDT-10_1038-S44319-024-00208-4.

## Disclosure and competing interests statement

The authors declare no competing interests.

# Expanded View Figures

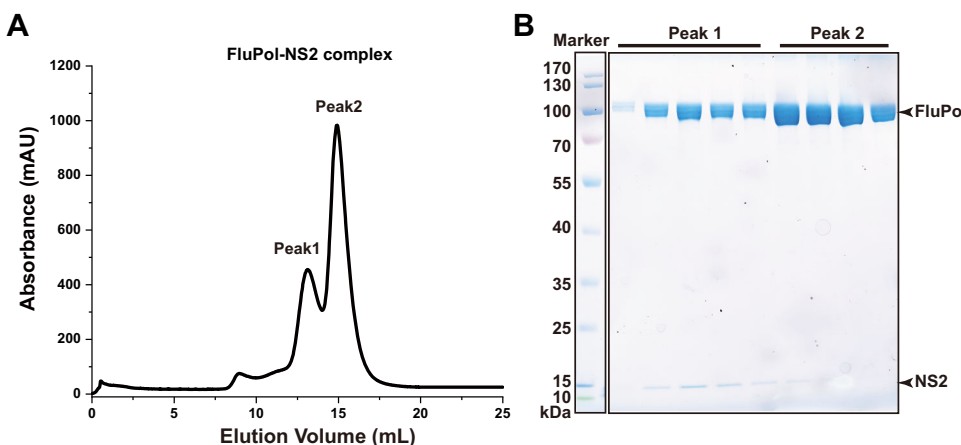

**Figure EV1.  Gel-filtration profile of the FluAPol-NS2 complex.**

(**A**) Size-exclusion chromatography of the FluAPol-NS2 complex. (**B**) SDS-PAGE profile of the FluAPol-NS2 complex.

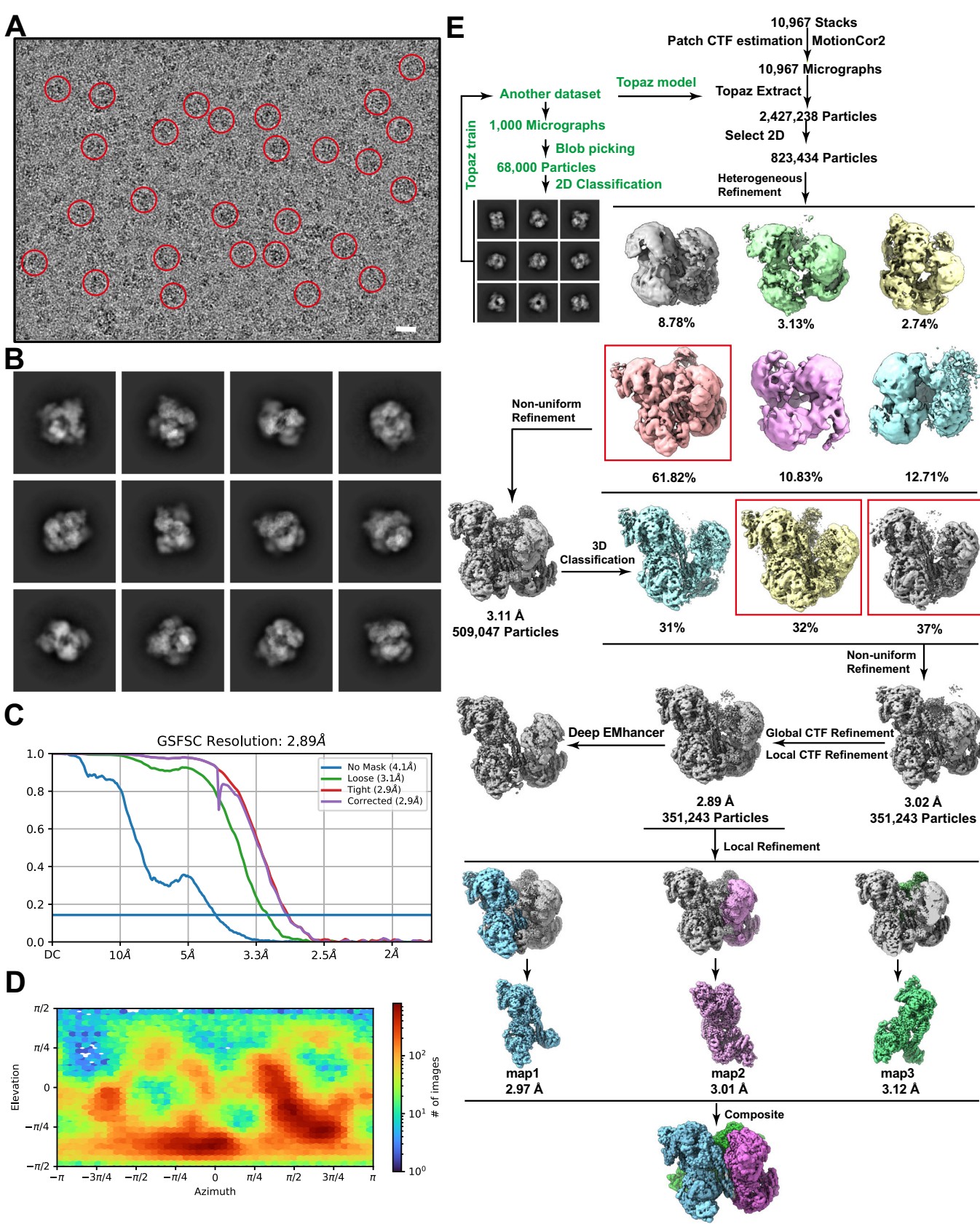

◀ **Figure EV2. Cryo-EM analysis of the FluAPol-NS2 complex.**

(A) A representative cryo-EM micrograph of the FluAPol-NS2 complex is shown. The complex particles are indicated by red circles. Scale bar: 20 nm. (B–E) Typical 2D class average images (B), FSC curves (C), Euler angle distribution (D), image processing workflow (E) of the FluAPol-NS2 complex.

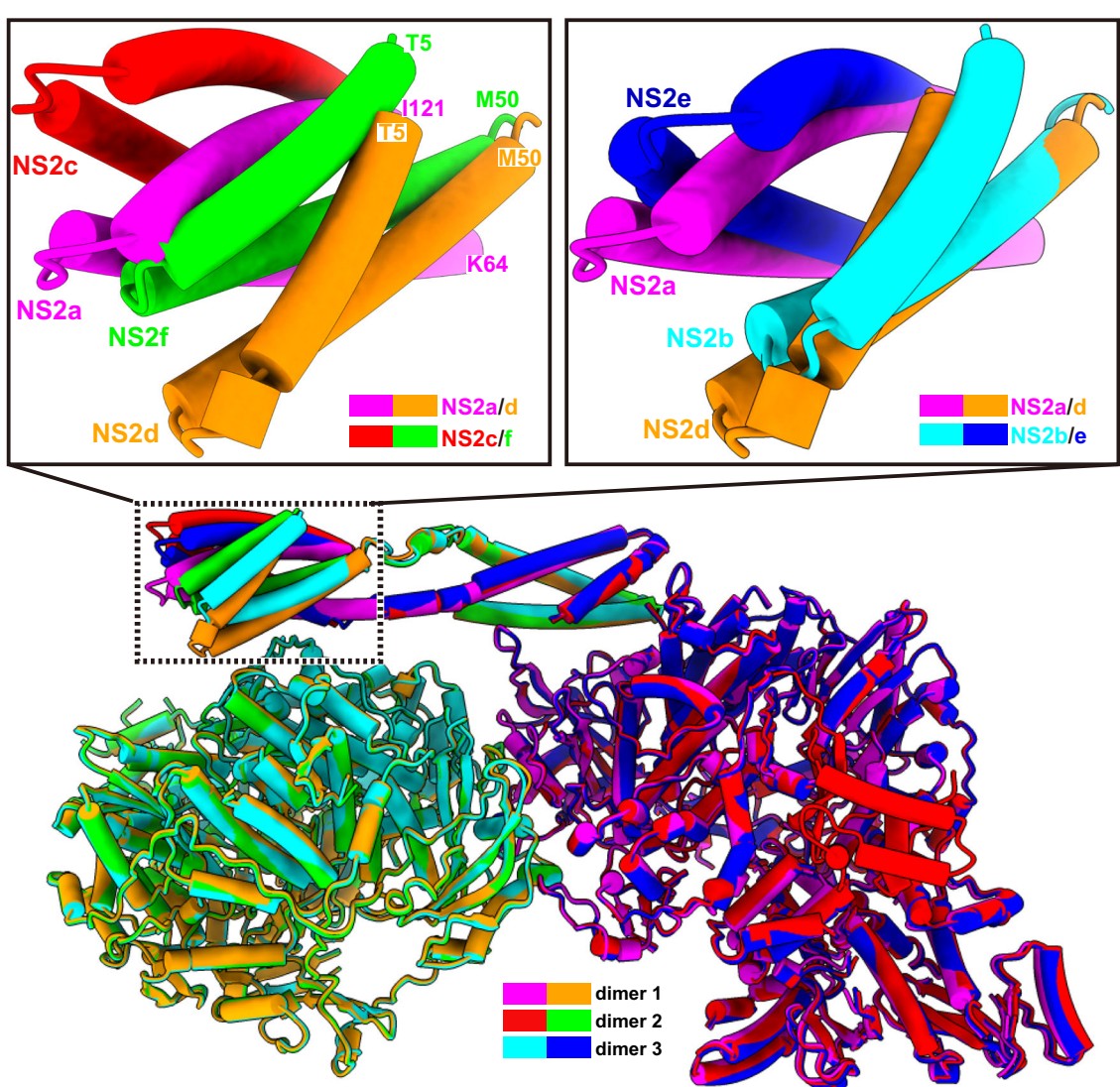

**Figure EV3. Flexibility of the domain-swapped NS2 dimers.**

The FluAPol-NS2 hexamer complex could be divided into three parts based on the EM density maps, each of them containing a FluAPol dimer and a domain-swapped NS2 dimer. We overlaid three parts, and three FluAPol dimers could be superimposed well. NS2 shows almost the same conformation at one end but significantly different conformation at the other end.

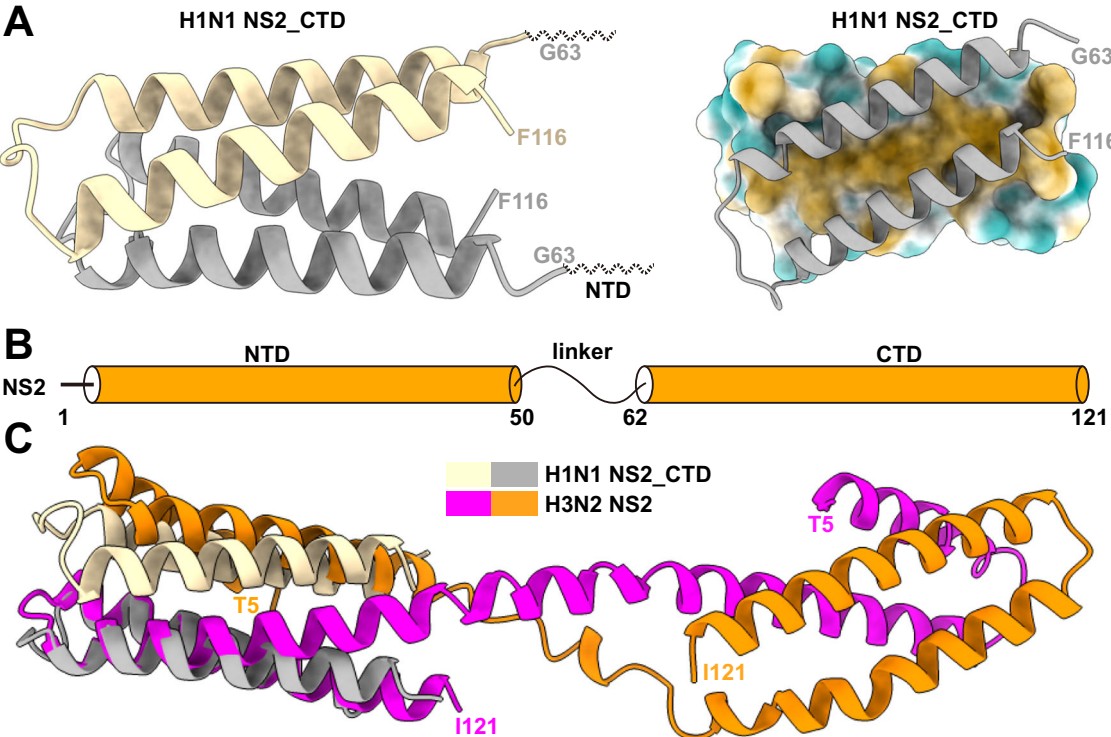

**Figure EV4. Structure of the full-length influenza NS2.**

(A) The crystal structure of the H1N1 NS2 CTD (PDB ID:1PD3) is shown. The NS2 CTD could form a dimer mainly through hydrophobic interactions. (B) Schematic diagram of the domains of the NS2 protein. (C) Structural overlay of the previously solved structure of the NS2 CTD dimer with the domain-swapped NS2 dimer solved.

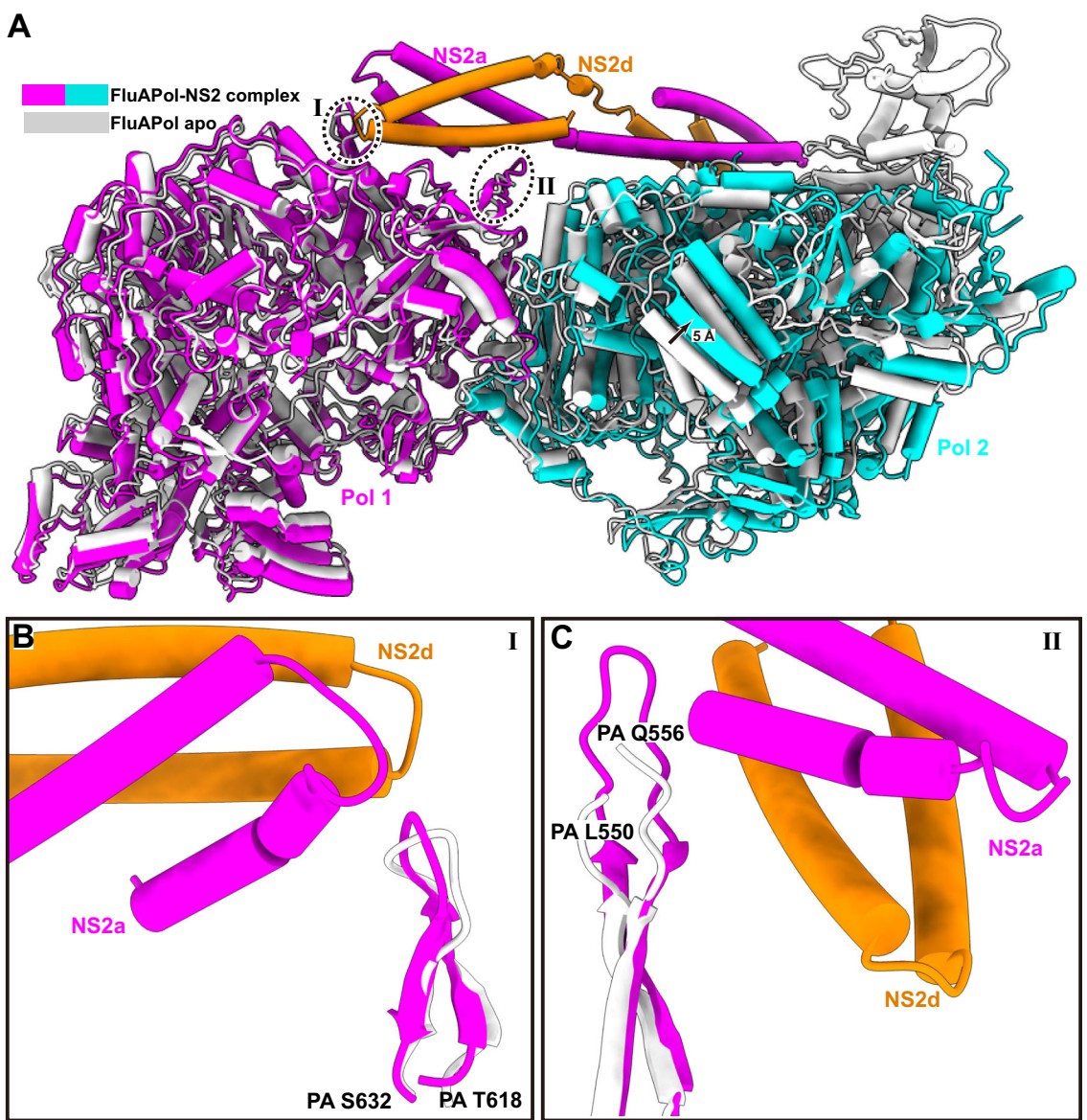

**Figure EV5. Conformational changes of the influenza polymerase dimer induced by NS2 binding.**

(A) A superimposition of structures of FluAPol-NS2 complex and apo FluAPol is shown. (B, C) Close up of site I (B) and II (C).

