## [Peer Review File · EMBO Reports]

NS2 induces an influenza A RNA polymerase hexamer and acts as a transcription to replication switch

Junqing Sun, Lu Kuai, Lei Zhang, Yufeng Xie, Yanfang Zhang, Yan Li, Qi Peng, Yuekun Shao, Qiuxian Yang, Wen-xia Tian, Junhao Zhu, Jianxun Qi, Yi Shi, Tao Deng, and George Gao

Corresponding author(s): George Gao (gaof@im.ac.cn), Yi Shi (shiyi@im.ac.cn), Tao Deng (dengt@im.ac.cn)

Review Timeline:

Submission Date:	14th Jun 24
Editorial Decision:	18th Jun 24
Revision Received:	3rd Jul 24
Accepted:	5th Jul 24

Editor: Achim Breiling

Transaction Report: A revised version of this manuscript was transferred to EMBO reports following peer review at the EMBO Journal.

Dear Prof. Gao,

Thank you for transferring your revised manuscript to EMBO reports. I now went through the manuscript, the referee reports from The EMBO Journal (attached again below) and your further point-by-point response (appeal letter).

Looking through the files, I would like to invite you to provide a final revised manuscript with the understanding that the remaining concerns of referee #2 must be addressed in the further revised manuscript and a final point-by-point response, as indicated in your appeal letter (by adding the new data mentioned in Addendum 1 and by discussing the related publications appropriately, in particular Zhang et al. J. Virol 2024 - PMID 38054704).

Moreover, I have these editorial requests.

- Please provide your final manuscript text file as .docx formatted file (including legends for main figures, EV figures and tables), but without the figures included. Figure legends should be compiled at the end of the manuscript text.

- Please upload individual production quality figure files as .eps, .tif, .jpg (one file per figure), of main figures and EV figures. Please upload these as separate, individual files upon re-submission.

- Please upload a completed author checklist (for EMBO reports as target journal), which you can download from our author guidelines (<https://www.embopress.org/page/journal/14693178/authorguide>). Please insert page numbers in the checklist to indicate where the requested information can be found in the manuscript.

- Please add 5 keywords to the manuscript text and order the manuscript sections like this, using these names: Title page - Abstract - Keywords - Introduction - Results - Discussion - Methods - Data availability section - Acknowledgements - Disclosure and Competing Interests Statement - References - Figure legends - Expanded View Figure legends

- Regarding data quantification and statistics, please make sure that the number "n" for how many independent experiments were performed, their nature (biological versus technical replicates), the bars and error bars (e.g. SEM, SD) and the test used to calculate p-values is indicated in the respective figure legends (also for EV figures and all those in an Appendix). Please also check that all the p-values are explained in the legend, and that these fit to those shown in the figure. Please provide statistical testing where applicable. Please avoid the phrase 'independent experiment', but clearly state if these were biological or technical replicates. Please also indicate (e.g. with n.s.) if testing was performed, but the differences are not significant. In case n=2, please show the data as separate datapoints without error bars and statistics. See also:

<http://www.embopress.org/page/journal/14693178/authorguide#statisticalanalysis>

If n<5, please show single datapoints for diagrams. Please add proper statistics to the diagrams shown in Figs. 2D/E, 3E, 5C, S5.

- Please add to each legend (main, EV and Appendix figures, where applicable) a 'Data Information' section explaining the statistics used or providing information regarding replicates and scales. See:

- Please add scale bars of similar style and thickness to microscopic images, using clearly visible black or white bars (depending on the background). Please place these in the lower right corner of the images themselves. Please do not write on or near the

bars in the image but define the size in the respective figure legend.

- Please make sure that all figure panels and Tables are called out separately and sequentially. Presently, separate callouts for Fig. 1A/B and for Appendix Table S1 seem missing. Please check.

- We now use CRediT to specify the contributions of each author in the journal submission system. CRediT replaces the author contribution section. Please use the free text box to provide more detailed descriptions and do NOT provide your final manuscript text file with an author contributions section. See also our guide to authors: <https://www.embopress.org/page/journal/14693178/authorguide#authorshipguidelines>

- Please make sure that all the funding information is also entered into the online submission system and is complete and similar to the one in the manuscript text file (in the Acknowledgements).

- Thanks for providing source data (SD). You have been contacted already by our source data coordinator, who indicated which figure panels we would need source data for. However, the SD you have provided does not match to the checklist. I attach again the source data checklist. Please check and make sure that all the requested source data is provided. Moreover, please provide all the numerical data for panels 2D, 2E, 3E, 4E and 5C. Please upload all source data for one figure as one folder ZIPed together. Finally, please upload the filled in source data checklist with your final revised files.

In addition, I would need from you:

- a short, two-sentence summary of the manuscript
- three to four short bullet points (two lines) that highlighting the key findings of your study
- a schematic summary figure (in jpeg or tiff format with the exact width of 550 pixels and a height of not more than 400 pixels) that can be used as a visual synopsis on our website.

I look forward to seeing a revised version of your manuscript when it is ready. Please let me know if you have questions or comments regarding the revision.

Kind regards,

Achim

Referee #1:

In their revision, the authors have presented a greatly improved manuscript. All points are carefully addressed. The illustrations now also leave little to be desired visually.

In particular, a PDB validation report has now been submitted, which indicates a careful and technically flawless structure determination. The manuscript should hence be put forward for publication.

Referee #2:

The experimental results are the structure of NS2 dimer and its interactions with the polymerase. However, the interpretation of NS2 functions by the authors is questionable.

The authors interpret the function of NS2 as regulating viral RNA synthesis, ignoring its essential function as the nuclear-export protein.

The interpretation of the mini-genome assays could be that when NS2 is not present, vRNA-RNP could not be exported out of the nucleus. The accumulation of vRNA-RNP in the nucleus results in increase of mRNA synthesis. When vRNA-RNP is exported by NS2, the level of mRNA is reduced. The assays reported in this manuscript do not separate RNPs in the cytoplasm from those in the nucleus. Transport by NS2 can change the accumulation of viral RNAs in different organelles and alter the viral RNA synthesis that only occurs in the nucleus.

All interpretations of NS2 functions must be based on solid experimental evidence. The simple mini-genome assays do not distinguish different mechanisms of function by NS2.

Dear Prof. Gao,

Thank you for transferring your revised manuscript to EMBO reports. I now went through the manuscript, the referee reports from The EMBO Journal (attached again below) and your further point-by-point response (appeal letter).

Looking through the files, I would like to invite you to provide a final revised manuscript with the understanding that the remaining concerns of referee #2 must be addressed in the further revised manuscript and a final point-by-point response, as indicated in your appeal letter (by adding the new data mentioned in Addendum 1 and by discussing the related publications appropriately, in particular Zhang et al. J. Virol 2024 - PMID 38054704).

Responses: We have addressed the remaining concerns of referee #2 in this point-by-point response letter. We have added Addendum 1 (in the appeal letter) in the manuscript as Appendix Figure S3 and discussed our recent related publications appropriately in Discussion (lines 345-348).

Moreover, I have these editorial requests.

- Please provide your final manuscript text file as .docx formatted file (including legends for main figures, EV figures and tables), but without the figures included. Figure legends should be compiled at the end of the manuscript text.

Responses: Completed.

- Please upload individual production quality figure files as .eps, .tif, .jpg (one file per figure), of main figures and EV figures. Please upload these as separate, individual files upon re-submission.

Responses: We have now provided as required.

Responses: They have been revised as required.

- Please upload a completed author checklist (for EMBO reports as target journal), which you can download from our author guidelines (<https://www.embopress.org/page/journal/14693178/authorguide>). Please insert page numbers in the checklist to indicate where the requested information can be found in the manuscript.

Responses: Completed.

- Please add 5 keywords to the manuscript text and order the manuscript sections like this, using these names:

Title page - Abstract - Keywords - Introduction - Results - Discussion - Methods - Data availability section - Acknowledgements - Disclosure and Competing Interests Statement - References - Figure legends - Expanded View Figure legends

Responses: Completed.

- Regarding data quantification and statistics, please make sure that the number "n" for how many independent experiments were performed, their nature (biological versus technical replicates), the bars and error bars (e.g. SEM, SD) and the test used to calculate p-values is indicated in the respective figure legends (also for EV figures and all those in an Appendix). Please also check that all the p-values are explained in the legend, and that these fit to those shown in the figure. Please provide statistical testing where applicable. Please avoid the phrase 'independent experiment', but clearly state if these were biological or technical replicates. Please also indicate (e.g. with n.s.) if testing was performed, but the differences are not significant. In case n=2, please show the data as separate datapoints without error bars and statistics. See also:

<http://www.embopress.org/page/journal/14693178/authorguide#statisticalanalysis>

If n<5, please show single datapoints for diagrams. Please add proper statistics to the diagrams shown in Figs. 2D/E, 3E, 5C, S5.

Responses: Completed.

- Please add to each legend (main, EV and Appendix figures, where applicable) a 'Data Information' section explaining the statistics used or providing information regarding replicates and scales. See:

Responses: Completed.

- Please add scale bars of similar style and thickness to microscopic images, using clearly visible black or white bars (depending on the background). Please place these in the lower right corner of the images themselves. Please do not write on or near the bars in the image but define the size in the respective figure legend.

Responses: Completed.

- Please make sure that all figure panels and Tables are called out separately and sequentially. Presently, separate callouts for Fig. 1A/B and for Appendix Table S1 seem missing. Please check.

Responses: We have checked and modified as suggested.

- We now use CRedit to specify the contributions of each author in the journal submission system. CRedit replaces the author contribution section. Please use the free text box to provide more detailed descriptions and do NOT provide your final manuscript text file with an author contributions section. See also our guide to authors:

<https://www.embopress.org/page/journal/14693178/authorguide#authorshipguidelines>

Responses: Completed

- Please make sure that all the funding information is also entered into the online submission system and is complete and similar to the one in the manuscript text file (in the Acknowledgements).

Responses: They are all right.

- Thanks for providing source data (SD). You have been contacted already by our source data coordinator, who indicated which figure panels we would need source data for. However, the SD you have provided does not match to the checklist. I attach again the source data checklist. Please check and make sure that all the requested source data is provided. Moreover, please provide all the numerical data for panels 2D, 2E, 3E, 4E and 5C. Please upload all source data for one figure as one folder ZIPed together. Finally, please upload the filled in source data checklist with your final revised files.

Responses: Completed.

In addition, I would need from you:

- a short, two-sentence summary of the manuscript
- three to four short bullet points (two lines) that highlighting the key findings of your study
- a schematic summary figure (in jpeg or tiff format with the exact width of 550 pixels and a height of not more than 400 pixels) that can be used as a visual synopsis on our website.

Responses: They have now been included as required.

I look forward to seeing a revised version of your manuscript when it is ready. Please let me know if you have questions or comments regarding the revision.

Please use this link to submit your revision: <https://embor.msubmit.net/cgi-bin/main.plex>

Kind regards,

Achim

Referee #1:

In their revision, the authors have presented a greatly improved manuscript. All points are carefully addressed. The illustrations now also leave little to be desired visually.

In particular, a PDB validation report has now been submitted, which indicates a careful and technically flawless structure determination. The manuscript should hence be put forward for publication.

Response: We thank the reviewer for the encouraging comments on our manuscript.

Referee #2:

The experimental results are the structure of NS2 dimer and its interactions with the polymerase. However, the interpretation of NS2 functions by the authors is questionable.

The authors interpret the function of NS2 as regulating viral RNA synthesis, ignoring its essential function as the nuclear-export protein.

The interpretation of the mini-genome assays could be that when NS2 is not present, vRNA-RNP could not be exported out of the nucleus. The accumulation of vRNA-RNP in the nucleus results in increase of mRNA synthesis. When vRNA-RNP is exported by NS2, the level of mRNA is reduced. The assays reported in this manuscript do not separate RNPs in the cytoplasm from those in the nucleus. Transport by NS2 can change the accumulation of viral RNAs in different organelles and alter the viral RNA synthesis that only occurs in the nucleus.

All interpretations of NS2 functions must be based on solid experimental evidence. The simple mini-genome assays do not distinguish different mechanisms of function by NS2.

Response: We are deeply sorry for our negligence in making this point clear in the original manuscript. According to the literature, the nuclear export of vRNP of influenza virus

requires forming a nuclear export complex of vRNP-M1-NS2 with host nuclear export protein Crm1 (e.g., Martin & Helenius, 1991, O'Neill *et al.*, 1998, Paterson & Fodor, 2012). To confirm this generally accepted conclusion, we further conducted an *in-situ* FISH experiment under our experimental conditions (Addendum 1). It directly showed that, NS2 alone could not exert its function on vRNP nuclear export unless the M1 protein is co-expressed in the system. We have now included this result as Appendix Figure S3 in the manuscript as a validation of our system in studying the regulatory role of NS2 on transcription and replication (lines 208-216). Moreover, we have also added a paragraph in Discussion section to further exclude the concern and to discuss the different amino acids identified in the NES1 of NS2 NTD in exerting its independent functions on viral RNA syntheses and vRNP nuclear export (lines 359-372).

Addendum 1. The vRNPs are restricted in the nucleus when NS2 is expressed alone.

(A) HEK-293T cells were transfected with plasmids as indicated for 36 h and then probed against NA vRNA and M1 mRNA using Cy5 and Cy3 labeled single-molecule inexpensive FISH (smiFISH) probes. DAPI marks the cellular nuclei. Mock, no transfection. -PB2, the expression plasmids of M1, NS2 and vRNP except for PB2 protein were co-transfected. vRNP, the expression plasmids of vRNP were co-transfected. vRNP+NS2, the expression plasmids of vRNP and NS2 proteins were co-transfected. vRNP+M1, the expression plasmids of vRNP and M1 proteins were co-transfected. vRNP+M1+NS2, the expression plasmids of vRNP, M1, and NS2 proteins were co-transfected. Scale-bar, 20 μm . (B)

Zoomed in view of the vRNP+NS2+M1 co-transfection group, as marked by the dashed rectangle in (A). Scale-bar, 5 μ m. (C) The efficiency of nuclear export of vRNP is determined by the ratio of vRNA in the cytosolic (Cyto%). (D) Quantification of the cytosolic fraction of vRNAs (Cyto%) of cells from the vRNP+NS2, the vRNP+M1, or the vRNP+M1+NS2 co-transfection groups.

References

Martin K, Helenius A (1991) Nuclear transport of influenza virus ribonucleoproteins: the viral matrix protein (M1) promotes export and inhibits import. *Cell* 67: 117-130

O'Neill RE, Talon J, Palese P (1998) The influenza virus NEP (NS2 protein) mediates the nuclear export of viral ribonucleoproteins. *Embo J* 17: 288-296

Paterson D, Fodor E (2012) Emerging roles for the influenza A virus nuclear export protein (NEP). *Plos Pathogens* 8: e1003019

Prof. George Gao
Institute of Microbiology, CAS
CAS Key Laboratory of Pathogen Microbiology and Immunology
No. 1 Beichen West Road, Chaoyang District
Beijing, Beijing 100101
China

Dear Prof. Gao,

I am very pleased to accept your manuscript for publication in the next available issue of EMBO reports. Thank you for your contribution to our journal.

Yours sincerely,
